# Structural Study of Nano-Clay and Its Effectiveness in Radiation Protection against X-rays

**DOI:** 10.3390/nano12142332

**Published:** 2022-07-07

**Authors:** Elfahem Sakher, Billel Smili, Mohamed Bououdina, Stefano Bellucci

**Affiliations:** 1Laboratory of Saharan Natural Resources, Faculty of Science and Technology, University of Adrar, National Highway No. 06, Adrar 01000, Algeria; 2Laboratory of Energy Environment and Information System (LEEIS), Department of Material Science, Faculty of Science and Technology, University of Adrar, National Highway No. 06. Adrar 01000, Algeria; billel.smili@univ-adrar.edu.dz; 3Department of Mathematics and Sciences, Faculty of Humanities and Sciences, Prince Sultan University, Riyadh 11586, Saudi Arabia; moboudina@gmail.com; 4INFN-Laboratori Nazionali di Frascati, Via E. Fermi 54, 00044 Frascati, Italy

**Keywords:** nano-clays, radiation shielding, radiation protection, X-ray diffraction, FTIR, Rietveld analysis

## Abstract

With the increasing applications of nuclear technology, radiation protection has become very important especially for the environment and the personnel close to radiation sources. Natural clays can be used potentially for shielding the X-ray radiations. In this study, the correlation between structural parameters and radiation shielding performance of natural clay extracted from Algerian Sahara (Adrar, Reggan, and Timimoune) was investigated. Phase composition and structural parameters (lattice parameters, average crystallite size, and microstrain) were determined by the Rietveld refinements of X-ray diffraction patterns in the frame of HighScore Plus software. The obtained results showed that the studied clays are nanocrystalline (nano-clay) since the calculated crystallite size was ≈3 nm for the feldspar phase. FTIR spectra confirmed the presence of all phases already detected by XRD analysis besides Biotite (around the band at 3558 cm^−1^). The remaining bands corresponded to absorbed and adsorbed water (3432 cm^−1^ and 1629 cm^−1^, respectively) and atmospheric CO_2_ (2356 cm^−1^). The shielding properties (mass absorption coefficient—µ/*ρ* and radiative attenuation rate—RA) for (green-yellow, green, and red) clays of Adrar, (red, white, and white-red) clays of Reggan, and red clay of Timimoune at same energy level were examined. The results of clay samples were compared with each other. The obtained results indicated that the green clay of Adrar exhibited the superior radiation shielding, i.e., 99.8% and 243.4 cm^2^/g for radiative attenuation rate and mass absorption coefficient, respectively.

## 1. Introduction

A few decades ago, rays were discovered and revolutionized many research fields for cutting-edge technologies. Nonetheless, their use presents serious risks that may outweigh their benefits. In this context, radioactive pollution is one of the most dangerous hazards threatening the environment and our planet including humans, animals, and plants.

Historically, in 1895 at the University of Wurzburg, Wilhelm Conrad Roentgen (1845–1923) [1,2] was studying various physical phenomena such as the cathode rays from an evacuated glass tube. When he observed a glowing fluorescent screen, Roentgen discovered the mysterious penetrating radiation called X-rays [2,3]. Nowadays, X-rays offer wide range of applications covering medical [4], military [5], space [6], security [7], and industrial [8] fields and in scientific research [9,10].

After that discovery, Henri Becquerel believed that phosphorescent uranium salts released rays after long exposure to the sun, but he abandoned this hypothesis and discovered in 1896 that uranium salts cause rays to occur naturally, meaning that the substance itself causes rays [11].

Marie Curie called this phenomenon radioactivity, even though Henri Becquerel [11] discovered it. She won the Nobel Prize twice alongside her husband Pierre and Henry Becquerel in physics and again in chemistry for discovering radium and polonium radioactive elements [12]. During World War I, She also contributed to the dissemination and development of mobile X-ray devices for armies [13].

Radiation is the energy released as electromagnetic waves or particles [14]. The sources of radiation include radon gas, the energy given off by a radioisotope, medical X-rays, and cosmic rays from outer space [15].

One can recall the use of radiation in many fields, for example, in medical applications (detection of SARS-CoV-2, radiation for cancer treatment, radio, and scans) [16,17], industrial fields [18], and military fields (radiological weapons, atomic and nuclear bombs rockets, mines, and ammunition) [19]. The utilization of radiation has pros and cons, and therefore, researchers have discovered several ways to reduce its negative effects, such as installations buried underground [20], storage in lead containers [21], a glass system (CaO-K_2_O-Na_2_O-P_2_O_5_ and Bi_2_O_3_–Na_2_O–TiO_2_–ZnO–TeO_2_) [22,23], nanomaterials [24], and clays and nano-clays [20,25].

Clays and nano-clays possess diverse interesting properties, including electrical [26,27], thermal [28,29], mechanical [30,31], and physicochemical [32,33]. Therefore, they are considered essential and important components in many areas such as cosmetics [34], cooking utensils [35], heavy metal filters [36], and geotechnical engineering (used to build roads, dams, mortar walls, airports, and landfills) [37].

The most commonly used and well-known materials for X-ray radiation shielding are lead (Pb) [38], tungsten (W) [39], and iron (Fe) [40]. Although they prevent exposure from radiation, toxicity and environmental aspects have pushed researchers toward the search for new less hazardous and cost-effective alternatives. This study aimed to investigate natural clays from the south of Algeria (Sahara) as potential candidates for radiation shielding. Particular emphasis was devoted to study the physicochemical properties of seven types of clays from three regions, namely, Adrar, Reggan, and Timimoune in correlation with their X-ray shielding ability including the radiative attenuation rate and mass absorption coefficient.

## 2. Experimental

The clay powders were examined by the X-ray diffraction (XRD) method using the X-ray ADRX Benchtop diffractometer (Ontario, Canada) equipped with a Cu-Kα radiation source (λ_Cu_ = 0.15418 nm). The analysis of phase composition was performed by using ICDD (PDF-2, 2014) and COD-2021 database files. Structural parameters were determined from the refinements of XRD patterns using the HighScore Plus program (version 3.0.4) (Almelo, The Netherlands) based on the Rietveld method [41].

The Fourier transform infrared spectra (FTIR) were recorded in the range 4000–400 cm^−1^ using the Agilent Cary 600 Series FTIR spectrometer (Springvale, Australia). A reference spectrum of KBr was subtracted to yield the final clay spectrum. The system software was capable of baseline correction, subtraction, and peak deconvolution of the measured spectra.

The clay samples were tested for X-ray shielding by using an X-ray apparatus (3B SCIENTIFIC PHYSICS (230 V, 50/60 Hz) 1000657) (Hamburg, Germany). The distance from the X-ray beam source to the clay was 5 cm. The samples were exposed to X-rays at the tube anode voltage of 30 kV with a switchable and electronically stabilized emission current in the range 0–80 μA. A Geiger-Muller (1000661) (Hamburg, Germany) tube was used to measure the X-ray transmission and a self-quenching Halogen trigger counter tube was used for registering X-radiation. The natural clay sample dimensions are given in Table 1. Each sample was exposed independently, and the mean value of each sample was calculated. The same procedure was also completed on synthetized clay samples with different thicknesses (5, 10, and 15 mm) in order to compare the shielding ability among the samples.

X-rays produced in X-ray tubes are also called Coolidge X-ray tubes [42] or hot cathode tubes. The principle is as follows: electrons emitted by a cathode (a filament, most often made of tungsten, heated by the passage of an electric current) are accelerated by a high potential difference in the direction of a target consisting of a metal anode (also made of tungsten) [43]. The different types of interactions between the X-ray beam and a material are:(i)Transmitted without changing direction [44].(ii)Transmitted by changing direction (at an angle) or diffused. The dissemination can be carried out without loss of energy (elastic scattering, it is at the origin of X-ray diffraction by crystals [42]) or with loss of energy (part of the energy is given up to an electron), and it is called inelastic diffusion which is at the origin of the Compton effect [45].(iii)Absorbed by atoms: under the action of incident radiation, an electron from an atom in the sample can be ejected from the electronic layer it occupied, which is the photoelectric effect, with the ejected electron being called a photoelectron. An electron from a higher layer can replace the ejected electron. This electronic jump is accompanied by an X-ray called fluorescence radiation [43,45].

The well-known Beer–Lambert law describes the absorption of X-rays by matter. When X-rays of intensity I pass through a substance of length dx, the X-ray intensity decreases by dI according to [46]:(1)dI=−μ I dx
where *µ* is the linear absorption coefficient and represents the characteristic of the material.

Without absorption of photons in the sample (*x* = 0), the incident intensity is *I*_0_. The integration of Equation (1) gives the following expression [47]:(2)$$I=I_{0}e^{-\mu x}$$

The linear absorption coefficient *µ* of X-rays by a material strongly depends on the nature of the atoms for a given frequency. Additionally, the absorption is more likely considered as the atomic number Z of the material [22,23]. Since the volumetric mass (*ρ*) of a substance reflects both the nature and quantity of the substance per unit volume, it would be more appropriate to relate the absorption coefficient to it (*ρ*) [48]:(3)$$I=I_{0}e^{-\mu x\;\rho x}$$
where *ρx* represents the mass thickness (the mass per unit area of a material layer of thickness *x*).

The following Equation (4) introduces the mass absorption coefficient *µ*/*ρ* [cm^2^/g]. It can also be written as follows [48]:(4)$$\frac{\mu }{\rho }=\frac{dI}{I}\frac{1}{\rho \;dx}$$

The radiative attenuation percentages *RA* (%) of materials are calculated using the following Equation (5) [49]:(5)$$RA=\left(\frac{I_{0}-I}{I_{0}}\right)\times 100$$

## 3. Result and Discussion

### 3.1. Structural Study

The recorded XRD pattens for the studied clay sample powders (green-yellow clay of Adrar (S1), green clay of Adrar (S2), red clay of Adrar (S3), red clay of Reggan (S4), white clay of Reggan (S5), white-red clay of Reggan (S6), and red clay of Timimoune (S7)) are shown in Figure 1. Each pattern was arbitrarily shifted vertically for better observation. Although there were differences in the XRD patterns of different samples, the following general statements can be highlighted:(i)The diffraction patterns displayed numerous peaks corresponding to different crystalline phases in each sample, where the most intense were present in (S6), (S5), and (S2).(ii)There was the presence of the same peak in all samples (2θ = 26.68°) but its intensity changed, which corresponded to the quartz phase (Figure 2).(iii)There was the presence of new peaks in samples (S6) (2θ = 27.54°) and (S1) (2θ = 11.60°) which were absent in the remaining samples.(iv)The peak located at (2θ = 12.33°) disappeared from sample (S2), while it was present in the remaining samples.

**Figure 1 nanomaterials-12-02332-f001:**
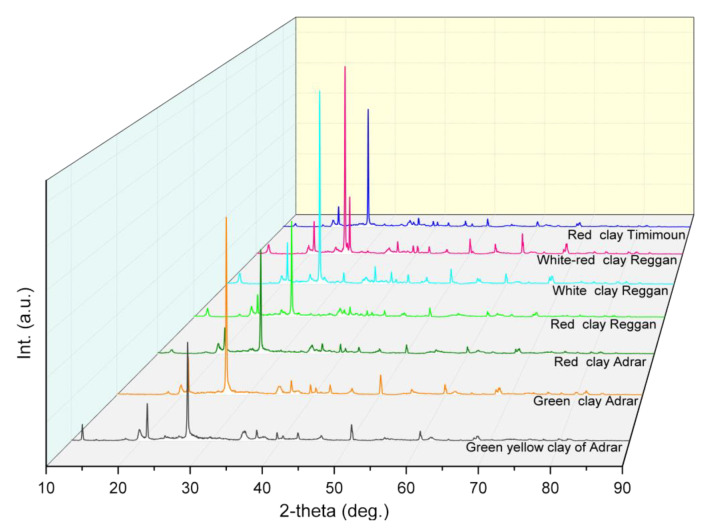
XRD patterns of the studied natural clay samples.

All the observed differences may be due to the presence of different phases in each sample; hence, Rietveld refinement was performed to identify the different phases and determine more precisely the phase composition alongside structural parameters for each phase (Figure 2).

**Figure 2 nanomaterials-12-02332-f002:**
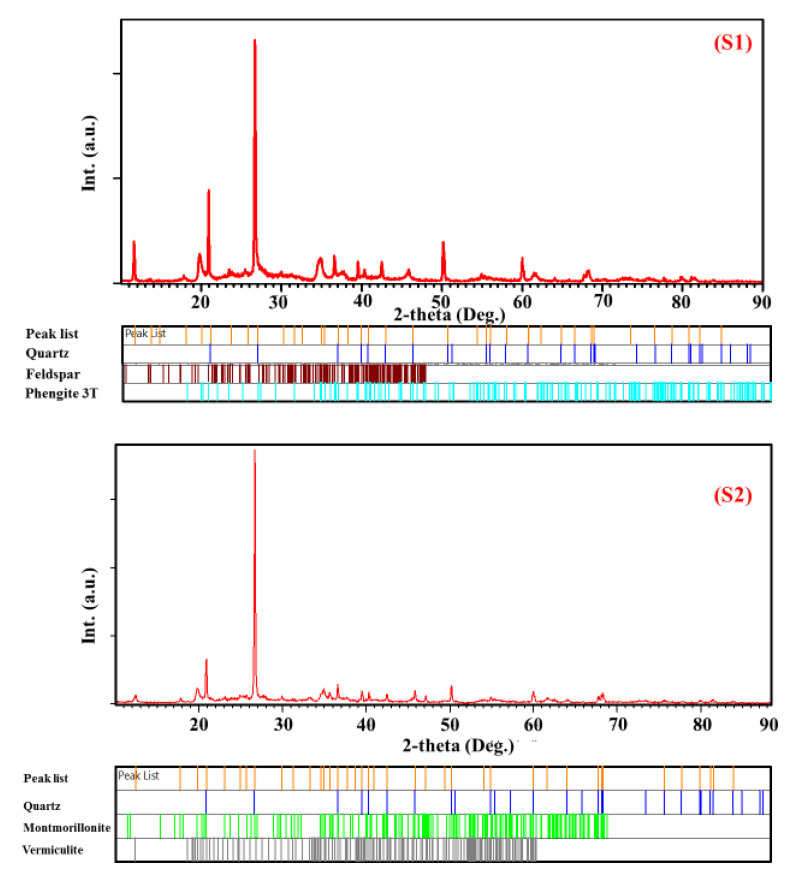
Rietveld refinements of XRD patterns of (S1) green-yellow clay—Adrar, (S2) green clay—Adrar, (S3) red clay—Adrar, (S4) red clay—Reggan, (S5) white clay—Reggan, (S6) white-red clay—Reggan, and (S7) red clay—Timimoune.

The Rietveld refinement by fitting the experimental X-ray diffraction patterns represents the ideal method for structural characterization [50], including qualitative and quantitative analyses of multiphase materials as well as different clay types [51]. Several researchers have adopted the XRD and Rietveld methods to determine the structural parameters of various systems [52,53,54,55].

Figure 2 displays the refined XRD patterns for (green-yellow, green, and red) clays of Adrar, (red, white, and white-red) clays of Reggan, and red clay of Timimoune based on the initial structural models presented in Table 2.

From the Rietveld analysis, it could be noted that the samples were completely different in terms of their constituent phases (Figure 2), which can be summarized as follows:-Quartz was present in all samples with different percentages;-Ph-3T was present in (S1) and (S3);-Mu was found in (S3), (S4), and (S7);-K was present in (S4), (S5), and (S6);-F was found only in (S1);-Mo and V were found in (S2);-S, N, and Ph were present in (S5), (S6), and (S7), respectively.

When using the HighScore Plus program to analyze the samples, it was shown that there were different phases (10 phases). Table 2 illustrates the structural characteristics of the primary phases used in the fitting of the experimental XRD patterns. These theoretical details used during the Rietveld refinements are mentioned for comparison, since many researchers have relied on the HighScore Plus program to analyze the XRD patterns of clays [56,57,58].

Figure 3 presents the quantitative analysis of the studied clay samples (S1, S2, S3, S4, S5, S6, and S7). For sample (S1), it was observed that the predominant phases were ph-3T and F with a ratio of 55.6% and 27.9%, respectively. It is also important to note that these two phases were found only in sample S1 (Figure 3 (S1)).

Furthermore, the (S2) sample was characterized by two phases that were found only in this sample (Figure 2 (S2)), namely, Ph with a small percentage of up to 3.3% and Mo (30.7%), whereas the predominant phase was the Q phase with a maximum ratio of 57%.

The Mu phase was clearly detected with a significant amount in samples (S3), (S4), and (S7) (Figure 3 (S3, S4, S7)), i.e., 27.2%, 50.3%, and 27.3%, respectively.

In Figure 3 (S6), the sample (S6) contained mainly three phases with the predominance of Ph phase, i.e., 45.6%, 25.5%, and 28.7% for Ph, K, and Q, respectively.

Herein, it is important to highlight that the Q phase participated in the formation of all samples, with a maximum amount in (S2) (57%) and the lowest amount in (S1) (16.5%).

The characterization of the structure of the clay powders was carried out by determining the cell parameters (a, b, c, α, β, γ, and unit cell volume), the crystallite size (nm), and the microstrain (%) by the Rietveld refinement. This method is based on fitting an experimental XRD pattern from a simulated crystallographic model that is as close as possible and depends on the analytical functions. All the results are reported in Table 3.

From Table 3, it can be observed that the smallest crystallite size value of ~3 nm was obtained for the F phase for sample (S1), whereas the maximum value of ~517 nm was obtained for the montmorillonite phase for sample (S2). For the Q phase, the smallest value for the crystallite size of ~73 nm was found for sample (S2) compared to all samples. For the variation of the microstrain, it is noted that the smallest value was obtained for the Q1 phase irrespective of the clay samples, i.e., setting in the range 0.077–0.178%, while the highest value of the microstrain belonged to the F phase, i.e., 7.6% for sample (S1). Furthermore, it is important to highlight that the values of crystallite size and microstrain are inversely proportional [59].

The smallest value of crystallite size in the nanometer scale was noted for sample (S3) for the phengite phase (~10 nm), which is comparatively slightly higher than its value (~3.2 nm) in sample (S1).

The muscovite phase dominant in S4 (Figure 3 (S4)) reaching about 50% was characterized by a crystallite size in the nanometer scale of ~7 nm. As for the kaolinite phase, it was characterized by similar values of a crystallite size of 9.6, 13.7, and 12.9 nm for samples (S4), (S5), and (S6), respectively.

The sanidine and nacrite phases were characterized by crystallite sizes equal to 21.0 nm and 11.1 nm, respectively, and they only appeared in samples (S5) and (S7).

### 3.2. FTIR Analysis

FTIR spectroscopy is an analytical characterization method with high sensitivity for the identification of substances’ components even at very small amounts, primarily bonding and functional groups [60]. This makes FTIR a very effective tool in mineralogy studies [51] and it was already applied for clay minerals [61]. Moreover, based on FTIR, one can identify crystalline [62], non-crystalline [63], and even organic [64] materials. The frequencies (cm^−1^) obtained from the recorded spectra in the region 4000–400 cm^−1^ of seven clay samples are displayed in Figure 4.

The infrared spectra of samples were arbitrarily shifted vertically to better depict the analysis. All spectra were characterized by bands that appeared in two well-defined spectral regions that originated from OH and SiO_4_ vibrations, which is supported by the data published in the literature [65,66,67]. Apparently, all examined samples presented twelve common bands located at ~470, ~531, ~701, ~787, ~908, ~1035, ~1112, ~1629, ~2356, ~3432, ~3625, and ~3696 cm^−1^. The tentative assignment for different bands in the vibrational spectra of the studied clay samples is presented in Table 4.

As seen from Figure 4 and the information listed in Table 4, all clay samples showed a broad absorption band at ~3432 cm^−1^ and a weak band at ~1629 cm^−1^ which could be assigned to the absorbed water [63]. The bands located around ~3432, ~3624, and ~3696 cm^−1^ were characteristic of OH group vibration elongation whereas the band ~2355 cm^−1^ could correspond to the presence of atmospheric CO_2_ in the samples [68]. The IR bands at ~1111 and ~1035 cm^−1^ were attributed to Si–O stretching out-of-plane and Si–O stretching in-plane, respectively, while the band at ~908 cm^−1^ suggested the presence of an Al–OH–Al functional group.

Further analysis of all IR spectra indicated the presence of Si–O and Si–O–Al stretching vibrations expressed by the absorption bands at ~786 and ~700 cm^−1^, respectively, while the latter band could also be a determinant of (Al, Mg)–O–H, Si–O–(Mg, Al) stretching.

The band located at ~530 cm^−1^ corresponded to the Si–O–Al stretching and Si–O bending. Finally, the broad absorption band around ~470 cm^−1^ was assigned to Si–O bending and Si–O–Fe stretching.

From XRD analysis, it was confirmed that all clay samples had different compositions of mineral phases, including quartz, feldspar, phengite-3T, montmorillonite, vermiculite, muscovite, kaolinite, sanidine, phengite, and nacrite. This corroborated with the FTIR analysis (Figure 5) by comparing the observed wavenumbers with the available data in the literature [63,69,70,71].

The presence of bands around ~1084, ~797, ~778, ~695, and ~468 cm^−1^ indicated that all clay samples contained the quartz phase. This supports the XRD results of clay samples (Figure 2).

For sample (S1), the IR spectrum revealed the presence of bands located at ~1031, ~778, ~694, ~528, ~468, and ~429 cm^−1^, which confirmed the presence of the feldspar phase in the red clay Reggan sample.

The muscovite phase existed in the samples (S3), (S4), and (S7) (Figure 2 (S3, S4, and S7)). Moreover, the IR spectrum confirmed this result; the muscovite characteristic bands were ~3619, ~3429, ~1629, ~1085, ~1031, ~1007, ~834, ~773, ~692, ~643, ~528, ~470, and ~425 cm^−1^.

The splitting bands in the region of 1000–4000 cm^−1^ including ~3726, ~3604, ~3430, and ~1029 cm^−1^ as well as the only band at ~466 cm^−1^ indicated the presence of the kaolinite phase in samples (S4), (S5), and (S6) (Figure 2 (S4), (S5), and (S6)). The observed absorption bands of kaolinite are in good agreement with findings of Golnaz et al. [63]. The characteristic bands corresponding to the sanidine phase were present at ~426, ~468, ~528, ~643, ~777, ~1029, and ~1156 cm^−1^.

The band at around ~528 cm^−1^ was attributed to the coupling between the K–O stretching vibration and the O–Si–O bending vibration. Moreover, the absorption band at ~643 cm^−1^ which was not present in the spectra of samples (S1), (S2), (S3), (S4), (S6), and (S7) corresponded to the O–Si (Al)–O bending vibrations. The characteristic montmorillonite absorption bands were observed at ~3619, ~3434, ~1029, ~912, and ~796 cm^−1^. Moreover, the vermiculite phase was defined by the following characteristic bands at ~3650, ~3409, and ~1635 cm^−1^.

As is clear in the IR spectra, the sample (S7) looked different compared to other samples. It was characterized by the presence of additional strong bands characteristic of the nacrite phase (Figure 5 (S7)), located at ~1111, ~1164, ~1007, ~970, ~910, ~798, ~753, ~694, ~533, ~541, ~473, and ~422 cm^−1^, as well as one band located at a higher wavenumber ~3625. Note that this phase did not appear except in sample (S7) (Figure 2 (S7)).

Herein, it is important to highlight that the FTIR analysis was very helpful in detecting some additional phases that were not apparent by XRD analysis. This includes biotite ~3557 cm^−1^ for samples (S4) and (S7) and kaolinite for samples (S2), (S3), and (S5). This may be because its percentage was very small (less than 2%), which is over the limit of X-ray detection [59].

### 3.3. Effect of Radiation Protection (Shielding)

From the obtained results presented in Table 5, it is noted that all the natural clay samples (extracted from the ground) exhibited good ability for radiation shielding, with a mean value above 99%. This may be due to the chemical elements and the constituent phases forming the investigated clay samples, irrespective of the nature of the clay (Figure 2).

The mass absorption coefficient (µ/*ρ*) is mainly affected by the nature and quantity of atoms that make up the substance as well as the thickness. Furthermore, it has been reported that the mass attenuation coefficient for a given material is dependent on the atomic number of the absorbing material and the incident photon energy. On the other hand, the exciting photon energy depends on material properties, including thickness. If the absorbing material consists of more than one element, the mass attenuation coefficient of the composite material is a function of the mass attenuation coefficients of individual elements and their respective mass fraction in the path of the photon beam [72]. It can be noticed that its value varied considerably, reaching the maximum value of 323 cm^2^/g for sample (S7) with (x = 5 mm) while the lowest value of 87.45 cm^2^/g was obtained for sample (S3) with (x = 15 mm). Indeed, the value of the mass absorption coefficient was found to be inversely proportional to the thickness, as reported by Ekinci N et al. [73].

Figure 6 depicts the variation of the radiative attenuation rate of different clay samples. It is noted that the obtained shielding parameters were remarkable, reaching up to 99.61% with a mass absorption coefficient of 263.89 cm^2^/g for sample (S2) and a thickness of 5 mm. These results are excellent compared to the results obtained by Akbulut S et al. [74], where the authors reported that the best radiative attenuation rate for the micronize clay–white cement mixture reached about 77%. Further, it is important to mention that the results obtained from the synthesized clays differed markedly from the natural clay samples. Indeed, the best sample for shielding was the manufactured (S2), and on the contrary, the weakest shielding for natural samples was also (S2). This may be because it is fragile, incoherent, and has very many cracks and holes, as found in nature [75]. The natural clay sample (S4), formed by very cohesive and hard components, demonstrated the best sample for shielding with the highest characteristics, i.e., 99.61% and 248.27 cm^2^/g.

The results of clay X-ray shielding were significantly high, reaching the best radiative attenuation rate of 99.79%, which is excellent compared to lead and tungsten [76,77]. These high shielding properties may be due to the components of the clay composed of various elements and metals with different contents, such as Al [78], Fe [79], Ti [80], Sr [81], and Si [82]. Consequently, the shielding rate values differed amongst the samples according to the proportions of the constituent elements.

As is clear, the effect of the thickness on the shielding was direct, as the radiative attenuation rate of materials increased with the increase in the thickness, as reported by Tsustumi K et al. [83].

The synthetized sample (S2) was the best clay sample with excellent ability to stop X-rays regardless of the change in thickness (Table 5). This may be due to the vermiculite phase [84] (Figure 4).

As for the comparison with the results obtained by Vagheian et al. [76], it should be highlighted that the authors used a weak energy in the range 8–14 KeV compared to the higher energy used in this study reaching up 30 keV.

In addition, Kim, S.-C. et al. [77] investigated tungsten-based materials with thicknesses in the range of millimeters subjected to different post-processing cold and hot rolling. It was found that the obtained results were preferable to lead-based materials, since the highest shielding rate of 100% was achieved for the hot-rolled plate with a thickness of 0.3 mm and an energy of 24.6 keV.

Furthermore, it is important to highlight that the clay material used in this study is natural eco-friendly, abundant, and very cheap compared to less abundant, expensive, and toxic metals such as lead or tungsten as reported in the literature. Furthermore, it does not matter if the vermiculite layer of shielding is thicker than the lead or tungsten layer to absorb the same amount of radiation because vermiculite has a very low density when compared to lead [84].

## 4. Conclusions

In this research, structural and X-ray shielding analyses were carried out on Algerian Sahara natural clays as potential alternatives to conventional metal or concrete materials. Phase identification and composition and structural parameters were determined by X-ray diffraction Rietveld refinements. Furthermore, the mass absorption coefficient (µ/*ρ*) and radiative attenuation ratio (RA) values of the studied clay samples were correlated with the phase composition and the thickness. The following conclusions could be highlighted:-XRD analysis showed that the studied and extracted clays from different regions of Adrar in Algeria could be classified as nanomaterials because the estimated crystallite size was found to be less than 100 nm, e.g., up to 3 nm for the feldspar phase.-The samples consisted of several phases, but the quartz phase was present irrespective of the nature of the clay with varying proportions. The vermiculite had the smallest ratio (3.3%), and the montmorillonite (30.7%) phase was found only in the sample green clay—Adrar.-FTIR bands appearing in two well-defined spectral regions originated from OH and SiO_4_ vibrations for all clay samples. FTIR confirmed the phases already observed by XRD analysis but also revealed new phases such as biotite (band ~3557 cm^−1^) for the samples red clay—Reggan and red clay—Timimoune as well as kaolinite (band ~466 cm^−1^) for the samples green clay—Adrar, red clay—Adrar, and white clay—Reggan.-The mass absorption coefficient (µ/*ρ*) and radiative attenuation ratio (RA) were found to be mainly affected by the nature, quantity of atoms, and thickness of the clay samples. The highest shielding values were obtained for sample green clay—Adrar, reaching 99.79% and 243.43 cm^2^/g for (RA) and (µ/*ρ*), respectively. This may be due to the vermiculite phase.

This research work demonstrated the promising potential of natural clays extracted from the region of Adrar in the Algerian Sahara as cost-effective alternatives to the conventional toxic heavy metals to preserve the environment and health.

## Figures and Tables

**Figure 3 nanomaterials-12-02332-f003:**
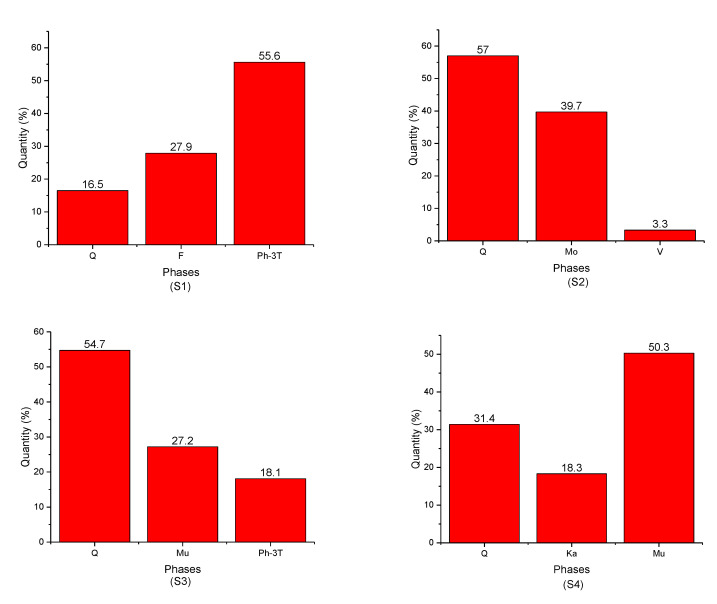
Evolution of the proportions of the identified phases as a function of clay samples: (S1) green-yellow clay—Adrar, (S2) green clay—Adrar, (S3) red clay—Adrar, (S4) red clay—Reggan, (S5) white clay—Reggan, (S6) white-red clay—Reggan, and (S7) red clay—Timimoune.

**Figure 4 nanomaterials-12-02332-f004:**
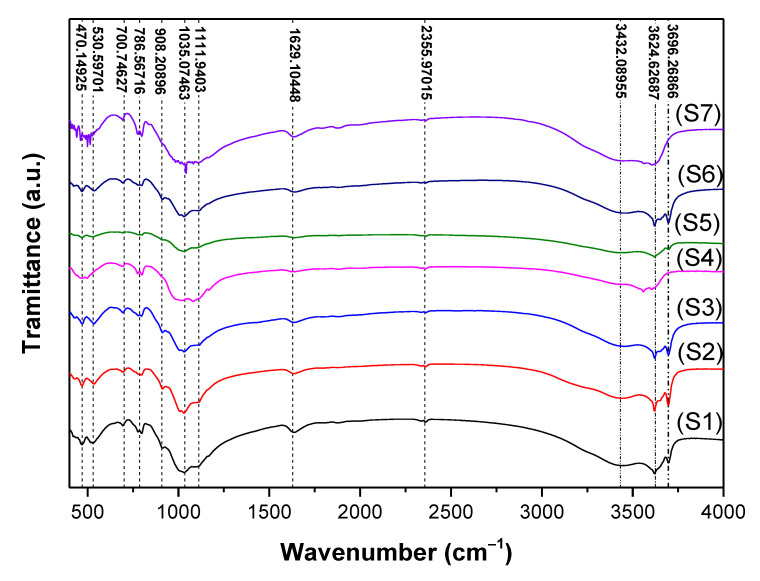
FTIR spectra of clay samples (S1, S2, S3, S4, S5, S6, and S7).

**Figure 5 nanomaterials-12-02332-f005:**
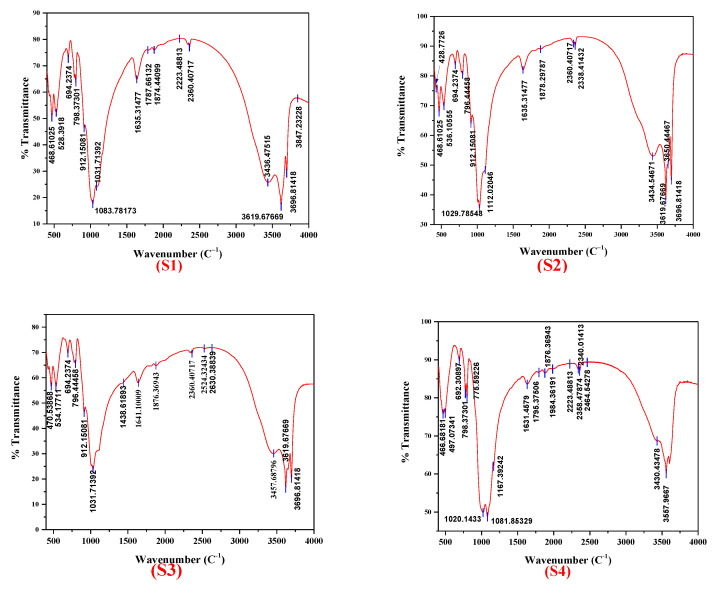
Evolution of stretching bands in the FTIR spectra of clay samples (S1, S2, S3, S4, S5, S6, and S7).

**Figure 6 nanomaterials-12-02332-f006:**
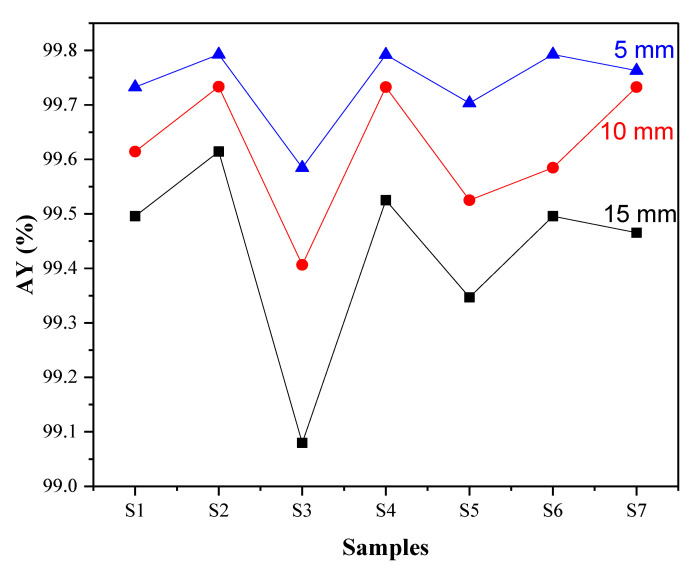
Radiative attenuation rate of the synthetized clay samples with different thicknesses (x = 5, 10, and 15 mm).

**Table 1 nanomaterials-12-02332-t001:** Characteristics and specifications of the studied samples.

Samples	Code	Color	Dimensions (mm)		Weight (g)		
				Natural Samples	Synthesized Samples		
				5 mm	X_1_ = 5 mm	X_2_ = 10 mm	X_3_ = 15 mm
Clay of Adrar	S1	Green-Yellow	Length = 30 Width = 25 Thickness: X_1_ = 5 X_2_ = 10 X_3_ = 15	7.23	12.48	17.10	30.33
	S2	Green		5.48	15.41	24.70	31.10
	S3	Red		/	9.70	18.41	30.17
Clay of Reggan	S4	Red		7.80	11.75	19.06	24.50
	S5	White		/	14.34	20.80	31.67
	S6	White-Red		7.15	13.11	21.54	33.14
Clay of Timimoune	S7	Red		7.87	10.35	18.19	28.90

**Table 2 nanomaterials-12-02332-t002:** Crystallographic parameters and properties of different phases used for the Rietveld refinements.

Phases	Chemical Formula	Crystal System	Space Group	Cell Parameters (a, b, c) in Å; (α, β, γ) in °		Volume (Å3)
Q	Si_3.00_O_6.00_	Hexagonal	P 3_1_ 2 1	a = 4.9160; b = 4.9160; c = 5.4090	α = 90; β = 90; γ = 120	113.21
Fr	Ca_0.80_Sr_3.20_Si_8.00_Al_8.00_O_32.00_	Triclinic	P-1	a = 8.3670; b = 9.1170; c = 9.2910	α = 89.57; β = 82.75; γ = 83.38	698.37
Ph	K_1.90_Na_0.10_Al_9.12_Mg_0.80_Fe_1.12_Si_12.96_O_47.84_F_0.16_H_7.84_	Monoclinic	C 1 2/c 1	a = 5.2230; b = 9.0620; c = 20.0440	α = 90; β = 95.74; γ = 90.00	938.93
Phe-3T	K_3.00_Al_5.36_Mg_1.92_Si_10.72_O_36.00_H_6.00_	Hexagonal	P 3_1_ 1 2	a = 5.2140; b = 5.2140; c = 29.7380	α = 90; β = 90; γ = 120	700.14
Mo	Al_4.00_Si_8.00_O_24.00_Ca_1.00_	Triclinic	P 1	a = 5.1800; b = 8.9800; c = 15.0000	α = 90; β = 90; γ = 90	697.75
V	Mg_13.64_Si_11.44_Al_4.56_O_48.00_	Monoclinic	C 1 2/c 1	a = 5.3490; b= 9.2550; c = 28.7217	α = 90; β = 93.5290; γ = 90	1419.17
Mu	K_2.91_Na_0.68_Ca_0.04_Al_11.01_Fe_0.12_Mg_0.09_Si_12.51_Ti_0.08_O_48.00_	Monoclinic	C 1 2/c 1	a = 5.1910; b = 9.0050; c = 20.1170	α = 90; β = 95.7730; γ = 90	933.43
K	Al_2.00_Si_2.00_O_9.00_H_4.00_	Triclinic	P 1	a = 5.1520; b = 5.1540; c = 7.3910	α = 74.954; β = 84.22; γ = 60.20	164.37
S	K_4.00_Si_12.00_Al_4.00_O_32.00_	Monoclinic	C 1 2/m 1	a = 8.6060; b = 13.0170; c = 7.1850	α = 90; β = 115.97; γ = 90	736.84
N	Al_8.00_Si_8.00_O_36.00_H_16.00_	Monoclinic	C 1 c 1	a = 8.9100 b = 5.1440; c = 14.5930	α = 90; β = 100.50; γ = 90	657.94

**
*Q (quartz), F (feldspar), Ph-3T (phengite-3T), Mo (montmorillonite), V (vermiculite), Mu (muscovite), K (kaolinite), S (sanidine), Ph (phengite), and N (nacrite).*
**

**Table 3 nanomaterials-12-02332-t003:** Structural parameters of clay samples (S1, S2, S3, S4, S5, S6, and S7) obtained from the Rietveld refinements.

Sample	Phases	Cell Parameters (a, b, c) in Å; (α, β, γ) in °		Volume (Å^3^)	Crystallite Size (nm)	Microstrain (%)
S1	Q	a = 4.9157; b = 4.9157; c = 5.4097	α = 90; β = 90; γ = 120	113.21	83.5	0.121
	F	a = 8.7663; b = 9.0274; c = 9.4109	α = 89.51; β = 82.32; γ = 83.14	732.78	3.0	7.611
	Ph-3T	a = 5.2568; b = 5.2568; c = 31.1166	α = 90; β = 90; γ = 120	744.68	3.2	7.109
S2	Q	a = 4.9162; b = 4.9162; c = 5.4065	α = 90; β = 90; γ = 120	113.16	73.6	0.178
	Mo	a = 5.2544; b = 8.94925; c =14.9868	α = 90; β = 90; γ = 90	704.73	516.7	4.422
	V	a = 5.36634; b = 9.16184; c = 28.7061	α = 90; β = 93.73347; γ = 90	1408.36	132.9	0.262
S3	Q	a = 4.91733; b = 4.91733; c = 5.4093	α = 90; β = 90; γ = 120	113.27	94.2	0.128
	Mu	a = 5.23369; b = 9.04311; c = 20.0579	α = 90; β = 95.57645; γ = 90	944.83	18.7	1.142
	Ph-3T	a = 5.2037; b = 5.2037; c = 29.6038	α = 90; β = 90; γ = 120	694.23	10.1	1.682
S4	Q	a = 4.9149; b = 4.9149; c = 5.4065	α= 90; β = 90; γ = 120	113.10	101.4	0.131
	K	a = 5.1465; b = 5.2105; c = 7.4517	α = 74.887; β = 84.49; γ = 60.07	167.05	9.6	2.215
	Mu	a = 5.2120; b = 9.0133; c = 20.0607	α = 90; β = 95.98; γ = 90	937.26	7.2	2.116
S5	Q	a = 4.91522; b = 4.9152; c = 5.4052	α = 90; β = 90; γ = 120	113.09	115.8	0.093
	K	a = 5.16362; b = 5.1861; c = 7.4264	α = 74.92; β = 84.37; γ = 60.07	166.29	13.4	1.748
	S	a = 8.6968; b = 12.9754; c = 7.1882	α = 90; β = 115.6405; γ = 90	731.27	21.0	0.972
S6	Q	a = 4.9149; b = 4.9149; c = 5.4067	α = 90; β = 90; γ = 120	113.11	130.8	0.077
	K	a = 5.1409; b = 5.1759; c = 7.4345	α = 74.88; β = 84.30; γ = 60.18	165.58	12.9	1.534
	Ph	a =5.2182; b = 9.0297; c = 20.0673	α = 90; β = 96.034; γ = 90	940.30	10.3	1.332
S7	Q	a = 4.9139; b = 4.9139; c = 5.4078	α = 90; β = 90; γ = 120	113.08	118.3	0.085
	Mu	a = 5.2337; b = 8.9646; c = 20.1286	α = 90; β = 95.36; γ = 90	940.26	11.2	1.223
	N	a = 8.9685; b = 5.1728; c = 14.4872	α = 90; β = 98.19908; γ = 90	665.23	11.9	1.874

**Table 4 nanomaterials-12-02332-t004:** Assignment of the bands in the powder infrared spectrum of clay samples.

Wavenumber (cm^−^^1^)	Assignment
470.1493	Si–O bending, Si–O–Fe stretching
530.5970	Si–O bending, Si–O–Al stretching
700.7463	Si–O stretching, Si–O–Al stretching
786.5672	Si–O stretching, Si–O–Al stretching, (Al, Mg)–O–H, Si–O–(Mg, Al) stretching
908.2090	Al_2_OH, AL–O–H vibrations
1035.0748	Si–O–Si, Si–O stretching
1111.9403	Si–O stretching(out-of-plane)
1629.1045	H–O–H stretching
2355.9702	Atmospheric CO_2_
3432.0896	Adsorbed water vibrations (H–O–H)
3624.6269	Inner OH groups, lying between the sheets of tetrahedral and octahedral units
3696.2687	Surface hydroxyls

**Table 5 nanomaterials-12-02332-t005:** Mass absorption coefficient (µ/*ρ*) and radiative attenuation percentages (RA) for natural and synthetized clays.

Thickness	Samples						
	S1	S2	S3	S4	S5	S6	S7
**Natural clays (V = 3.0 × 2.5 × 0.5 = 3.75 × 10 mm)**							
**Thickness x = 5 mm**							
**RA (%)**	99.52	99.38	/	99.61	/	99.44	99.55
**µ/*ρ* (cm^2^/g)**	217.39	218.05	/	248.27	/	184.93	212.96
**Synthetized clays**							
**Thickness x = 5 mm**							
**µ/*ρ* (cm^2^/g)**	248.98	263.89	174.79	309.01	167.12	237.02	323.06
**Thickness x = 10 mm**							
**µ *ρ* (cm^2^/g)**	237.81	243.43	143.31	308.31	167.07	228.45	283.13
**Thickness x = 15 mm**							
**µ/*ρ* (cm^2^/g)**	193.75	214.27	87.45	280.91	158.69	175.37	229.39

## Data Availability

Not applicable.

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
