# Peer review of "Structural Study of Nano-Clay and Its Effectiveness in Radiation Protection against X-rays"

_nanomaterials, 2022, doi:10.3390/nano12142332_

Round 1

Reviewer 1 Report

This paper is rising a very important aspect linked to radiation protection, referring to the clays that can be used for shielding the X-ray radiation. In this study, structural and microstructural parameters and radiation shielding performances of natural clay extracted from Algerian Sahara (Adrar, Reggan and Timimoune) are investigated.

The paper wanted to be a very extensive one presenting the  Rietveld refinements of X-ray diffraction patterns, phase composition, and structural/microstructural parameters such as lattice parameters, average crystallite size, and microstrain, and by Fourier transform infrared spectroscopy (FTIR). The paper is important and well structured, and the XRD and FTIR parts are well detailed and useful to readers in the field.

However, the paper has incomplete parts (eg missing bibliography, only marked on the first page), and the figures are unclear and difficult to follow.

I suggest the authors correct the mentioned aspects and resubmit the work for publication.

Author Response

Response to Reviewer # 1 Comments

This paper is rising a very important aspect linked to radiation protection, referring to the clays that can be used for shielding the X-ray radiation. In this study, structural and microstructural parameters and radiation shielding performances of natural clay extracted from Algerian Sahara (Adrar, Reggan and Timimoune) are investigated.

The paper wanted to be a very extensive one presenting the  Rietveld refinements of X-ray diffraction patterns, phase composition, and structural/microstructural parameters such as lattice parameters, average crystallite size, and microstrain, and by Fourier transform infrared spectroscopy (FTIR). The paper is important and well structured, and the XRD and FTIR parts are well detailed and useful to readers in the field.

However, the paper has incomplete parts (eg missing bibliography, only marked on the first page), and the figures are unclear and difficult to follow.

The authors are very thankful for the reviewers positive feedback on our manuscript.

The comments raised by the Reviewers are answered carefully point by point here after, and all changes and corrections are incorporated in the revised manuscript in red color.

Furthermore, the English has been improved signficantly and the revised manuscript has been proofread before resubmssion.

Comment # 1 - Incomplete parts (eg missing bibliography, only marked on the first page)

Response

We thank you for your efforts, for references, we have added references to the mentioned points like folow:

Historically, at the University of Wurzburg, on November 8, 1895, Wilhelm Conrad Roentgen (1845-1923) [1, 2] was studying various physical phenomena. He was studying the cathode rays from an evacuated glass tube. When he observed a glowing fluorescent screen, Roentgen discovered the mysterious penetrating radiation called X-rays [2,3.], and this discovery had a great impact in many fields, such as medical [4], military [5], space [6], security [7], industrial [8], and in research field [9, 10].

         [4] Zhao, T.; Zhang, S.-X. X-ray Image Enhancement Based on Nonsubsampled Shearlet Transform and Gradient Domain Guided Filtering. Sensors 2022, 22, 4074.

         [5]Thalhammer, S.; Hörner, A.; Küß, M.; Eberle, S.; Pantle, F.; Wixforth, A.; Nagel, W. GaN Heterostructures as Innovative X-ray Imaging Sensors—Change of Paradigm. Micromachines2022, 13, 147.

         [6]Parkman, T.; Nevrkla, M.; Jančárek, A.; Turňová, J.; Pánek, D.; Vrbová, M. Table-Top Water-Window Microscope Using a Capillary Discharge Plasma Source with Spatial Resolution 75 nm.  Sci.2020, 10, 6373.

         [7]Hassan, T.; Shafay, M.; Akçay, S.; Khan, S.; Bennamoun, M.; Damiani, E.; Werghi, N. Meta-Transfer Learning Driven Tensor-Shot Detector for the Autonomous Localization and Recognition of Concealed Baggage Threats. Sensors2020, 20, 6450.

        [8]Schmoeller, M.; Stadter, C.; Kick, M.K.; Geiger, C.; Zaeh, M.F. A Novel Approach to the Holistic 3D Characterization of Weld Seams—Paving the Way for Deep Learning-Based Process Monitoring. Materials2021, 14, 6928.

Comment # 2 - The figures are unclear and difficult to follow.

Response

The figures embedded in the manuscript have a lower resolution, principally when the MS Word document is converted into PDF file. However, we uploaded high resolution figures saved as image (png) during submission as separate files

Reviewer 2 Report

Authors analyzed several clays by X-ray diffraction and infrared spectroscopy, and revealed their compositions regarding the constituent mineral phases, as well as obtained information on the characteristic crystallite sizes by using Rietveld crystal structure refinement. In the second part of the paper, authors apply these materials for testing as X-ray radiation protection shield, and determine mass adsorption coefficients, and the shielding efficiency in the percent scale, in the function of the shield material thickness.

At certain materials thickness, authors obtain attenuation percentage above 99.79%, and call it excellent, comparing with similar values obtained in an earlier study by Vagheian et al. for metal lead. However in that study, samples of micron thicknesses were used, while in the present study, centimeter thick samples were used. Hence, such comparison is obviously erroneous, and makes no sense.

Unfortunately, most of the present findings are irrelevant, and the paper lacks the minimal scientific approach to interpret the experimental findings. Thus, no any relation is analyzed, or conclusion given, regarding what is the reason of different shielding performance of the different clays, which is probably the first and most interesting question a researcher would ask. In spite of the many hours used for collecting the experimental data, the basic numerical analysis of the attenuation coefficients for the given elemental compositions was not done.

Interestingly, only one idea is suggested for the stronger attenuation of one of the samples, which contains vermiculite phase. Authors for some reason include in this phase the element Astatine (At), which is rather strange, as it is radioactive and unstable with very short half-life. Very likely some mistake happened at some point of this research.

In general, the paper is not suitable for journal Nanomaterials because of its very low quality.

Author Response

Response to Reviewer 2 Comments

Authors analyzed several clays by X-ray diffraction and infrared spectroscopy, and revealed their compositions regarding the constituent mineral phases, as well as obtained information on the characteristic crystallite sizes by using Rietveld crystal structure refinement. In the second part of the paper, authors apply these materials for testing as X-ray radiation protection shield, and determine mass adsorption coefficients, and the shielding efficiency in the percent scale, in the function of the shield material thickness.

At certain materials thickness, authors obtain attenuation percentage above 99.79%, and call it excellent, comparing with similar values obtained in an earlier study by Vagheian et al. for metal lead. However in that study, samples of micron thicknesses were used, while in the present study, centimeter thick samples were used. Hence, such comparison is obviously erroneous, and makes no sense.

Unfortunately, most of the present findings are irrelevant, and the paper lacks the minimal scientific approach to interpret the experimental findings. Thus, no any relation is analyzed, or conclusion given, regarding what is the reason of different shielding performance of the different clays, which is probably the first and most interesting question a researcher would ask. In spite of the many hours used for collecting the experimental data, the basic numerical analysis of the attenuation coefficients for the given elemental compositions was not done.

Interestingly, only one idea is suggested for the stronger attenuation of one of the samples, which contains vermiculite phase. Authors for some reason include in this phase the element Astatine (At), which is rather strange, as it is radioactive and unstable with very short half-life. Very likely some mistake happened at some point of this research.

The authors are very thankful for the Reviewer pertinents comments, which will strengthen the discussion of the obtained results and improve the quality of the manuscript. All questions are considered seriousely and hereafetr we reprot our response to all Reviewer comments point by point, then all changes are incorporated in the revised manuscript in red. The revised version of the manuscript has been throughly revided for English to reach the standard of “Nanomaterials” and proofread before submission.

Point 1

At certain materials thickness, authors obtain attenuation percentage above 99.79%, and call it excellent, comparing with similar values obtained in an earlier study by Vagheian et al. for metal lead. However in that study, samples of micron thicknesses were used, while in the present study, centimeter thick samples were used. Hence, such comparison is obviously erroneous, and makes no sense.

Response

We thank the Reviewer for highlighting this intringing comment. As for the comparison made with the results obtained by Vagheian et al. [1], it should be highlighted that the authors used a weak energy in the range 8 - 14 KeV, compared to higher energy used in this study reaching up 30 keV.

In addition, Kim, S.-C. et al. [2] investigated tungsten-based materials with thicknesses in the range of millimeters subjected to different post processing cold and hot rolling. It was found that the obtained results are preferable to lead-based materials, since the highest shielding rate of 100% was achieved for the hot rolled plate with a thickness of 0.3 mm and an energy 24.6 keV.  

This reference was added with the correposning explanation, to support the results obtained in this study.

Furthermore, it is important to highlight that the clay material used in this study is natural eco-friendly, abundant, and very cheap compared to less abundant, expensive, and toxic metals such as lead or tengesten as reported in the literature. Also, it  doesn’t  matter  even thought  the  vermiculite  layer  of  shielding  should  be  thicker  than  lead or tungsten  layer  to absorb the same amount of radiation, because the vermiculite has very low density when compared to lead [3].

References

[1] Vagheian M, Sardari D, Saramad S, Ochbelagh DR. Experimental and theoretical investigation into X-ray shielding properties of thin lead films. International Journal of Radiation Research. 18(2) (2020) 263-74.

[2] Kim, S.-C. Improving the X-ray Shielding Performance of Tungsten Thin-Film Plates Manufactured Using the Rolling Technology. Appl. Sci. 11 (2021) 9111.

[3] Gülbiçim, H, Tufan, MC, Türkan, MN. The investigation of vermiculite as an alternating shielding material for gamma rays. Radiation Physics and Chemistry 130 (2017) 112-117.

Point 2

Unfortunately, most of the present findings are irrelevant, and the paper lacks the minimal scientific approach to interpret the experimental findings. Thus, no any relation is analyzed, or conclusion given, regarding what is the reason of different shielding performance of the different clays, which is probably the first and most interesting question a researcher would ask. In spite of the many hours used for collecting the experimental data, the basic numerical analysis of the attenuation coefficients for the given elemental compositions was not done.

Response

We are thenakful for the Reviewer insightful comment. The following statement has been added to the the manuscript:

“These high shielding properties may be due to the components of the clay composed of various elements and metals with different content, such as Al [4], Fe [5], Ti [6], Sr [7], and Si [8]. Consequently, the shielding rate values differ between the samples according to the proportions of the constituent elements.”

About the basic numerical analysis of the attenuation coefficients for the given elemental compositions: in fact, we have conducted only an experimental study for shielding and calculated the attenuation with structural and microstructural studies. In this regard, a correlation with the phase composition and the shielding rate can be drawn. Nevertheless, in our future research, we will certainly consider to add a theoretical study and compare it with the obtained results.

References

  • Manjunatha, HC, et al. Gamma, X-ray and neutron shielding parameters for the Al-based glassy alloys. Applied Radiation and Isotopes 139 (2018) 187-194.
  • Mengge, D., et al. "Using iron concentrate in Liaoning Province, China, to prepare material for X-Ray shielding. Journal of Cleaner Production 210 (2019) 653-659.
  • Fujisaki, K., et al. Effect of Skin and Subcutaneous Tissue on X-Ray Strain Measurement of Ti Implant." World Congress on Medical Physics and Biomedical Engineering, September 7-12, 2009, Munich, Germany. Springer, Berlin, Heidelberg, 2009.
  • Oğuzhan, Ö, et al. High rate X-ray radiation shielding ability of cement-based composites incorporating strontium sulfate (SrSO4) minerals." Kerntechnik 87.1 (2022) 115-124.
  • Manjunatha, HC, et al. A study of X-ray, gamma and neutron shielding parameters in Si-alloys." Radiation Physics and Chemistry 165 (2019) 108414.

Point 3

Interestingly, only one idea is suggested for the stronger attenuation of one of the samples, which contains vermiculite phase. Authors for some reason include in this phase the element Astatine (At), which is rather strange, as it is radioactive and unstable with very short half-life. Very likely some mistake happened at some point of this research.

Response

We appreciate the Reviewer pertiment comment. Indeed, we have re-analyzed the sample 2 by the Rietved method, where we found that the Astatine element has disappeared. This may be due to its instability, and the reason for its presence in the sample may be through French nuclear tests carried out in the Adrar region [9].

Reference

[9] Regnault, JM. France's search for nuclear test sites, 1957-1963. The journal of military history 67.4 (2003) 1223-1248.

Reviewer 3 Report

In this article, several types of clays were examined to determine the effectiveness of protection against radiation with their help. Structural and phase analysis can be assessed as qualitative. Evaluation of the efficiency of radiation attenuation is also beyond doubt. However, I have significant comments on the interpretation of the data obtained and the presentation of the results. Despite the comments, the article has good potential. If comments and recommendations are taken into account, the article may be published in Nanomaterials. Thus, my decision is major revision.

Comments

My main comment relates to the lack of a causal relationship between the phase composition, structure and radiation attenuation efficiency of clays. The main achievement in this article is a comparative analysis of several types of clays. This approach is not scientific enough. This work is technical report in present form. the authors need to do a deeper analysis of the experimental data and find out the reason why green clay of Adrar is more effective in protecting against radiation

Are authors sure this article is suitable for Nanomaterials? It should be confirmed that clays are indeed nanomaterials. The XRD data helps to calculate the size of the crystallite, or the size of the coherent scattering region, which may be nanoscale. However, if we argue like the authors (that if the crystallite size is less than 100 nm by X-ray), then almost all materials are nanomaterials, even, for example, steel. Studies using scanning electron microscopy will allow you to evaluate the microstructure.

 It is well known that the density of the material is the most important parameter for radiation shields [10.1016/j.radphyschem.2021.109556, 10.3390/nano12101642]. In this study, there are no even approximate density estimates. This can be done using XRD data, such as in https://doi.org/10.1016/j.ceramint.2021.09.064 and 10.4028/www.scientific.net/SSP.299.100. Perhaps the density results will help shed light on the reason for the higher efficiency of green clay of Adrar.

The labels in Figures 2, 3, 5 are too small to be legible. They must be enlarged.

English needs to be tested and improved. There are many mistakes.

Author Response

Response to Reviewer # 3 Comments

In this article, several types of clays were examined to determine the effectiveness of protection against radiation with their help. Structural and phase analysis can be assessed as qualitative. Evaluation of the efficiency of radiation attenuation is also beyond doubt. However, I have significant comments on the interpretation of the data obtained and the presentation of the results. Despite the comments, the article has good potential. If comments and recommendations are taken into account, the article may be published in Nanomaterials. Thus, my decision is major revision.

Comments

My main comment relates to the lack of a causal relationship between the phase composition, structure and radiation attenuation efficiency of clays. The main achievement in this article is a comparative analysis of several types of clays. This approach is not scientific enough. This work is technical report in present form. the authors need to do a deeper analysis of the experimental data and find out the reason why green clay of Adrar is more effective in protecting against radiation

 Are authors sure this article is suitable for Nanomaterials? It should be confirmed that clays are indeed nanomaterials. The XRD data helps to calculate the size of the crystallite, or the size of the coherent scattering region, which may be nanoscale. However, if we argue like the authors (that if the crystallite size is less than 100 nm by X-ray), then almost all materials are nanomaterials, even, for example, steel. Studies using scanning electron microscopy will allow you to evaluate the microstructure.

It is well known that the density of the material is the most important parameter for radiation shields [10.1016/j.radphyschem.2021.109556, 10.3390/nano12101642]. In this study, there are no even approximate density estimates. This can be done using XRD data, such as in https://doi.org/10.1016/j.ceramint.2021.09.064 and 10.4028/www.scientific.net/SSP.299.100. Perhaps the density results will help shed light on the reason for the higher efficiency of green clay of Adrar.

The labels in Figures 2, 3, 5 are too small to be legible. They must be enlarged.

English needs to be tested and improved. There are many mistakes.

The authors are very thankful for the reviewers positive feedback on our manuscript.

The comments raised by the Reviewer are answered carefully point by point hereafter, and all changes and corrections are incorporated in the revised manuscript in green.

Furthermore, the English has been improved signficantly and the revised manuscript has been proofread before resubmssion.

Comment# 1 - Are authors sure this article is suitable for Nanomaterials? It should be confirmed that clays are indeed nanomaterials. The XRD data helps to calculate the size of the crystallite, or the size of the coherent scattering region, which may be nanoscale. However, if we argue like the authors (that if the crystallite size is less than 100 nm by X-ray), then almost all materials are nanomaterials, even, for example, steel. Studies using scanning electron microscopy will allow you to evaluate the microstructure.

Response

We thank you for your this insightful remark. Nanomaterials are classified into families, such as nanopaticles and nanocrystals, and the difference between them is that the nanoparticles are of nanoscale and may consist of several nanocrystals or one crystal, but nanocrystals may be a component of particles exceeding even a micron.

Jaison Jeevanandam, in their review entitled "Review on nanoparticles and nanostructured materials:

history, sources, toxicity and regulations" (https://doi.org/10.3762/bjnano.9.98), define nanomaterials in detail as follows.

A nanometer (nm) is an International System of Units (Système international d'unités, SI) unit that represents 10−9meter in l ength. In principle, NMs are described as materials with length of 1–100 or 1000 nm in at least one dimension; however, they are commonly defined to be of diameter in the range of 1 to 100 nm.Today, there are several pieces of legislation in the European Union (EU) and USA with specific references to NMs. However, a single internationally accepted definition for NMs does not exist. Different organizations have a different in opinion in defining NMs [1]. According to the Environmental Protection Agency (EPA), “NMs can exhibit unique properties dissimilar than the equivalent chemical compound in a larger dimension” [2]. The US Food and Drug Administration (USFDA) also refers to NMs as “materials that have at least one dimension in the range of approximately 1 to 100 nm and exhibit dimension-dependent phenomena” [3]. Similarly, The International Organization for Standardization (ISO) has described NMs as a “material with any external nanoscale dimension or having internal nanoscale surface structure” [4]. Nanofibers, nanoplates, nanowires, quantum dots and other related terms have been defined based on this ISO definition [5]. Likewise, the term nanomaterial is described as “a manufactured or natural material that possesses unbound, aggregated or agglomerated particles where external dimensions are between 1 and 100 nm size range”, according to the EU Commission [6].

Recently, the British Standards Institution [7] proposed the following definitions for the scientific terms that have been used:

  • Nanoscale: Approximately 1 to 1000 or 100 nm size range.
  • Nanoscience: The science and study of matter at the nanoscale that deals with understanding their size and structure-dependent properties and compares the emergence of individual atoms or molecules or bulk material related differences.
  • Nanotechnology: Manipulation and control of matter on the nanoscale dimension by using scientific knowledge of various industrial and biomedical applications.
  • Nanomaterial: Material with any internal or external

structures on the nanoscale dimension.

  • Nano-object: Material that possesses one or more peripheral nanoscale dimensions.
  • Nanoparticle: Nano-object with three external nanoscale dimensions. The terms nanorod or nanoplate are employed, instead of nanoparticle (NP) when the longest and the shortest axes lengths of a nano-object are different.
  • Nanofiber: When two similar exterior nanoscale dimensions and a third larger dimension are present in a nanomaterial, it is referred to as nanofiber.
  • Nanocomposite: Multiphase structure with at least one phase on the nanoscale dimension.
  • Nanostructure: Composition of interconnected constituent parts in the nanoscale region.
  • Nanostructured materials: Materials containing internal or surface nanostructure.

We can also mention several articles that study nanoclay and were published in the nanomaterials journal:

  • https://doi.org/10.3390/nano12020177
  • https://doi.org/10.3390/nano9070917
  • https://doi.org/10.3390/nano11112789
  • https://doi.org/10.3390/nano3030550

We can also find in the following articles (in nanomaterials jornal), that the particles size are from the micron degree, but the crystallite size is from the nano level, which are materials classified as materials with nano properties

  • https://doi.org/10.3390/nano12122086
  • https://doi.org/10.3390/nano12101635
  • https://doi.org/10.3390/nano12060938

As for scanning electron microscopy imaging, we certainly agree with the ereviewer comment. However, at our University we don’t any SEM to perform such mesurements. But certainly, we will plan to carry out morphological observations by SEM along EDS for chemical composition.

Comment # 2 -It is well known that the density of the material is the most important parameter for radiation shields [10.1016/j.radphyschem.2021.109556, 10.3390/nano12101642]. In this study, there are no even approximate density estimates. This can be done using XRD data, such as in https://doi.org/10.1016/j.ceramint.2021.09.064 and 10.4028/www.scientific.net/SSP.299.100. Perhaps the density results will help shed light on the reason for the higher efficiency of green clay of Adrar.

Response

We have included the efficiency values when calculatingthe mass absorption coefficient µ/ρ [cm²/g]

The linear absorption coefficient µ of X-rays by a material strongly depends on the nature of the atoms for a given frequency. Also, the absorption is more likely considered as the atomic number Z of the material. Since the volumetric mass (ρ) of a substance reflects both the nature and quantity of the substance per unit volume, it would be more appropriate to relate the absorption coefficient to it (ρ) [40]:

                                    (3)

where ρxrepresents the mass thickness (the mass per unit area of a material layer of thickness x).

The following 4 equation (4) introduces the mass absorption coefficient µ/ρ [cm²/g]. It can also be written as follow [40]:

                                                      (4)

We thank you for suggesting this important reference, and we have added the suggested reference to support the article

  • The utilization of radiation has pros and cons, so researchers have discovered several ways to reduce its negative effects, such as installations buried underground [20], storage in lead containers [21],glass system(CaO-K2O-Na2O-P2O5 and Bi2O3–Na2O–TiO2–ZnO–TeO2) [22, 23], nanomaterials[24] and clays / nano-clays [20, 25].
  • The linear absorption coefficient µ of X-rays by a material strongly depends on the nature of the atoms for a given frequency. Also, the absorption is more likely considered as the atomic number Z of the material [22, 23].

Comment # 3 - The figures are unclear and difficult to follow (The labels in Figures 2, 3, 5 are too small to be legible. They must be enlarged).

Response

The figures embedded in the manuscript have a lower resolution, principally when the MS Word document is converted into PDF file. However, we uploaded high resolution figures saved as image (png) during submission as separate files

Reviewer 4 Report

The paper Structural and microstructural Study of Nano-clay and its effectiveness in radiation protection (X-rays) is devoted to investigation of the natural clay extracted from Algerian Sahara. XRD, FTIR techniques were applied for the samples characterization. Shielding properties against X-rays were investigated. The topic of this paper is critically actual especially in area of shielding materials. The work is of scientific interest to the audience of the Nanomaterials. The data are reliable and do not cause much doubt. Nevertheless, there are several points before the paper can be published. I hope that authors after major revisions can improve the paper and can publish it in Nanomaterials.

  1. First of all I recommend designing the paper in accordance with the Nanomaterials template. The references aren’t also designed in accordance with the guidelines.
  2. The title of the paper I recommend to correct to “Structural Study of Nano-clay and its effectiveness in radiation protection against X-rays”
  3. The manuscript does not contain the information about microstructure. The data about microstructure features obtained using SEM, for example, is absent. So it is wrong to say that you studied the microstructure. If you decide to leave the microstructure, so please, include the microscopy investigations.
  4. The Introduction part must be improved with new relevant literature about nanomaterials for radiation shielding. I suggest using relevant literature [please see and discuss:

https://doi.org/10.3390/ma14143772;

https://doi.org/10.1016/j.vacuum.2018.07.017;

https://doi.org/10.1016/j.radphyschem.2021.109556;

https://doi.org/10.1016/j.surfcoat.2016.10.061].

  1. Table with samples specification is needed.
  2. Figures 2 and 5 do not read. The resolution is poor, it should be increased.
  3. There is no information about nanomaterials only few words in Conclusion. Since the paper was submitted to Nanomaterials journal, this point must be clarified.
  4. English must be improved, because now there are some typos and grammatical errors.

Authors must explain some details and improve the paper in accordance with my comments. The paper should be sent to me for the second analysis after the major revisions.

Author Response

Response to Reviewer # 4 Comments

The paper Structural and microstructural Study of Nano-clay and its effectiveness in radiation protection (X-rays) is devoted to investigation of the natural clay extracted from Algerian Sahara. XRD, FTIR techniques were applied for the samples characterization. Shielding properties against X-rays were investigated. The topic of this paper is critically actual especially in area of shielding materials. The work is of scientific interest to the audience of the Nanomaterials. The data are reliable and do not cause much doubt. Nevertheless, there are several points before the paper can be published. I hope that authors after major revisions can improve the paper and can publish it in Nanomaterials.

First of all I recommend designing the paper in accordance with the Nanomaterials template. The references aren’t also designed in accordance with the guidelines.

Response

We thank you for your effors in reviewing our paper. We also appreciate the reviewer recommending designing the paper and references in accordance with the Nanomaterials template. Indeed, the paper and references were now formatted according to the nanomaterials template.

The title of the paper I recommend to correct to “Structural Study of Nano-clay and its effectiveness in radiation protection against X-rays”

Response

We thank the reviewer for this important note, and we have modified the name to “Structural Study of Nano-clay and its effectiveness in radiation protection against X-rays”

The manuscript does not contain the information about microstructure. The data about microstructure features obtained using SEM, for example, is absent. So it is wrong to say that you studied the microstructure. If you decide to leave the microstructure, so please, include the microscopy investigations.

Response

As for the electron microscope, unfortunately we cannot perform it at this time, because we don’t have such expensive equipment in our university. Nevertheless, we will take this insightful remark into consideration in the future research and we will do our best to conduct SEM observationd alongside EDS analysis elsewhere.

The Introduction part must be improved with new relevant literature about nanomaterials for radiation shielding. I suggest using relevant literature [please see and discuss:

https://doi.org/10.3390/ma14143772;

https://doi.org/10.1016/j.vacuum.2018.07.017;

https://doi.org/10.1016/j.radphyschem.2021.109556;

https://doi.org/10.1016/j.surfcoat.2016.10.061].

Response

We appreciate the reviewer suggestion to add new references. Indeed, the references related to the research work have been added as follow:

  • The utilization of radiation has pros and cons, so researchers have discovered several ways to reduce its negative effects, such as installations buried underground [20], storage in lead containers [21], glass system(CaO-K2O-Na2O-P2O5 and Bi2O3–Na2O–TiO2–ZnO–TeO2) [22, 23], nanomaterials [24] and clays / nano-clays [20, 25].
  • The linear absorption coefficient µ of X-rays by a material strongly depends on the nature of the atoms for a given frequency. Also, the absorption is more likely considered as the atomic number Z of the material [22, 23].
  • Several researchers have adopted the XRD and Rietveld method to determine the structural parameters of various systems [52, 53, 54, 55].

[22] Sayyed, M. I., Albarzan, B., Almuqrin, A. H., El-Khatib, A. M., Kumar, A., Tishkevich, D. I., ... & Elsafi, M. (2021). Experimental and theoretical study of radiation shielding features of CaO-K2O-Na2O-P2O5 glass systems. Materials14(14), 3772.

[23] Sayyed, M. I., Askin, A., Zaid, M. H. M., Olukotun, S. F., Khandaker, M. U., Tishkevich, D. I., & Bradley, D. A. (2021). Radiation shielding and mechanical properties of Bi2O3–Na2O–TiO2–ZnO–TeO2 glass system. Radiation Physics and Chemistry186, 109556.

[52] Warcholinski, B., Gilewicz, A., Lupicka, O., Kuprin, A. S., Tolmachova, G. N., Ovcharenko, V. D., ... & Chizhik, S. A. (2017). Structure of CrON coatings formed in vacuum arc plasma fluxes. Surface and Coatings Technology309, 920-930.

[52] Warcholinski, B., Gilewicz, A., Kuprin, A. S., Tolmachova, G. N., Ovcharenko, V. D., Kuznetsova, T. A., ... & Chizhik, S. A. (2018). Mechanical properties of Cr-ON coatings deposited by cathodic arc evaporation. Vacuum156, 97-107.

Table with samples specification is needed.

Response

We have added a table containing characteristics and specifications of the studied samples

Table 1. Characteristics and specifications of the studied samples

Samples

Code

Color

Dimensions

(mm)

Weight

(g)

Natural Samples

Synthesized Samples

5 mm

X1=5 mm

X2=10 mm

X3=15 mm

Clay of Adrar

S1

Geen-Yellow

Length=30

Width=25

Thickness:

X1=5

X2=10

X3=15

7.23

12.48

17.10

30.33

S2

Green

5.48

15.41

24.70

31.10

S3

Red

/

9.70

18.41

30.17

Clay of Reggan

S4

Red

7.80

11.75

19.06

24.50

S5

White

/

14.34

20.80

31.67

S6

White-Red

7.15

13.11

21.54

33.14

Clay of Timimon

S7

Red

7.87

10.35

18.19

28.90

Figures 2 and 5 do not read. The resolution is poor, it should be increased.

Response

The figures embedded in the manuscript have a lower resolution, principally when the MS Word document is converted into PDF file. However, we uploaded high resolution figures saved as image (png) during submission as separate files.

There is no information about nanomaterials only few words in Conclusion. Since the paper was submitted to Nanomaterials journal, this point must be clarified.

Response

We have added several points focusing on nanomaterials and shielding, as well as references related to the properties of nanoclays.

The utilization of radiation has pros and cons, so researchers have discovered several ways to reduce its negative effects, such as installations buried underground [20], storage in lead containers [21], glass system(CaO-K2O-Na2O-P2O5 and Bi2O3–Na2O–TiO2–ZnO–TeO2) [22, 23], nanomaterials [24], and clays / nano-clays [20, 25].

Clays / nano-clays possess diverse interesting properties, such as electrical [26, 27], thermal [28, 29], mechanical [30, 31], and physicochemical [32, 33.].

English must be improved, because now there are some typos and grammatical errors.

Response

The English has been improved signficantly and the revised manuscript has been proofread before resubmssion.

Round 2

Author Response

Response to Reviewer # 2 Comments

The authors are very thankful for the Reviewer pertinents comments, which will strengthen the discussion of the obtained results and improve the quality of the manuscript. All questions are considered seriousely and hereafetr we reprot our response to all Reviewer comments point by point, then all changes are incorporated in the revised manuscript in red. The revised version of the manuscript has been throughly revided for English to reach the standard of “Nanomaterials” and proofread before submission.

We have chosen “Nanomaterials”journal due to its excellent reputation and high quality published papers, but we suspect that the reviewer has deleted part of our answer to the first point of his comment, and we hope that this is purely a mistake and not intentional, especially since most of his comments are negative and are lackingobjectivity and impartiality.

Point 1

At certain materials thickness, authors obtain attenuation percentage above 99.79%, and call it excellent, comparing with similar values obtained in an earlier study by Vagheian et al. for metal lead. However in that study, samples of micron thicknesses were used, while in the present study, centimeter thick samples were used. Hence, such comparison is obviously erroneous, and makes no sense.

Response

We thank the Reviewer for highlighting this intriguing comment. As for the comparison made with the results obtained by Vagheian et al. [1], it should be highlighted that the authors used a low energy,in the range 8 - 14 KeV, compared to higher energy used in this study, reaching up 30 keV.

In addition, Kim, S.-C. et al. [2]investigated tungsten-based materialswith thicknesses in the range of millimeters subjected to different post processing cold and hot rolling. It was found that the obtained results are preferable to lead-based materials, since the highest shielding rate of100% was achieved for the hot rolled plate with a thickness of 0.3 mm and an energy 24.6keV.

This reference was added with the correposning explanation, to support the results obtained in this study.

Reviewer’s answer to authors’ response 1.

Basically,intheirresponseauthorsadmitthatthereisnopointincomparisonof shieldingefficiencyoftheclaymaterialswiththeothercited worksdealingwith metallic plates. Which is fine. It follows, that the results of the present study are not comparabletothecitedliterature,thereforethecitedliteratureanditsdiscussion doesnotsupporttheresultsofthepresentstudy,contrarytothe statementofthe authors.

Response

It is important to highlight that the clay material used in this study is natural, eco-friendly, abundant, and very cheap, compared to less abundant, expensive, and toxic metals such as lead or tungesten, as reported in the literature.

Gülbiçim, Het al., made a comparison as follows “itdoesn’tmattereven thoughtthevermiculitelayerofshieldingshouldbethickerthanlead or tungstenlayerto absorb the same amount of radiation, because the vermiculite has very low density when compared to lead [3].”

References

[3] Gülbiçim, H, Tufan, MC, Türkan, MN. The investigation of vermiculite as an alternating shielding material for gamma rays.Radiation Physics and Chemistry 130 (2017) 112-117.

Point 2

Unfortunately, most of the present findings are irrelevant, and the paper lacks the minimal scientific approach to interpret the experimental findings. Thus, no any relation is analyzed, or conclusion given, regarding what is the reason of different shielding performance of the different clays, which is probably the first and most interesting question a researcher would ask. In spite of the many hours used for collecting the experimental data, the basic numerical analysis of the attenuation coefficients for the given elemental compositions was not done.

Response

We are thankful for the Reviewer insightful comment. The following statement has been added to the the manuscript:

“These high shielding properties may be due to the components of the clay composed of various elements and metals with different content, such as Al [4], Fe [5], Ti [6], Sr [7], and Si [8].Consequently, the shielding rate values differ amongst the samples, according to the proportions of the constituent elements.”

About the basic numerical analysis of the attenuation coefficients for the given elemental compositions: in fact, we have conducted only an experimental study for shielding and calculated the attenuation with structural and microstructuralstudies. In this regard, a correlation with the phase composition and the shielding rate can be drawn. Nevertheless, in our future research, we will certainly consider to adda theoretical study and compare it with the obtained results.

Reviewer’s answer to authors’ response 2

The newly added sentences just repeat the self-evident fact, that the shielding rate of differentmaterialsdependontheproportionoftheconstituentelements.Thereis nothing new in this. If this is the main finding of the authors, then it is not a scientific work,butatechnicalpaper,inwhichshieldingofvariousmaterialsare compared, withoutareasonableinterpretationoftheresults.Suchreportdoesnotmeetthe requirements ofNanomaterials.The work contains also IR and XRD characterization of the materials, but these results have no relation to the x-ray attenuation results.

Authorscalltheirresults“correlation”betweenphasecompositionandx-ray attenuation.However,thereisnoanycorrelation showninthis paper,justalistof attenuation values. of different materials of different thicknesses. Authors use incorrectly the word correlation.

Response

We have made great efforts to conduct this research, and it is important to focus on that the shielding is 99 %, and this may be an opportunity to open a new field of future research in which researchers invest to benefit humanity.

Most researchers agree that the main influencing parameter for shielding is the atomic number of the constituent elements, how they are related and their density. Therefore, the details mentioned in the FTIR and XRD of the studied samples give us many insightful information about the effect of the shielding strength.

Point 3

Interestingly, only one idea is suggested for the stronger attenuation of one of the samples, which contains vermiculite phase. Authors for some reason include in this phase the element Astatine (At), which is rather strange, as it is radioactive and unstable with very short half-life. Very likely some mistake happened at some point of this research.

Response

We appreciate the Reviewer pertiment comment. Indeed, we have re-analyzed the sample 2 by the Rietved method, where we found that the Astatine element has disappeared. This may be due to its instability, and the reason for its presence in the sample may be through French nuclear tests carried out in the Adrar region [9].

Reviewer’s answer to authors’ response 3

Thisishighlyunlikelythatanalysesrepeatedwithinaoneyearperiodcouldbe differentfortheexistence/disappearanceoftheAstanitephase.Nuclearradiation, whichaccordingtoauthors,mightinducethiselement,weredoneseveraldecades ago, a short living radioactive isotope should have disappeared by now or by the time of the first xrd experiment. Or if not, then it is not a short living isotope and it is still present.

Mostlikely,authorsmadeamistakeinthefirstRietveldfitting.Authorsresponse aboutthe possibleorigin ofAstaniteinthefirst versionofthemanuscript suggests their poor understanding of basics of radiation physics.

Response

Scientific research is based on experience and repetition of the experiment, and some errors may occur during the analysis, and the recognition of the error correction is positive and important in scientific research, and after your observations, we have re-analyzed the sample, as technology now allows re-analysis of samples by rays any time and in a very short period of time, that may take a few minutes. This is what helped us to re-analyze it again after receiving your comments and benefitting from them.

We think that your comments, Sir, have some contradictions. During your first comment, you confirm that the Astatine is unstable and has a short life “the element Astatine (At), which is rather strange, as it is radioactive and unstable with very short half-life”. But in the second comment, you wonder how it disappeared ?! As for the weak understanding, we hope that you will contact us to develop beekeepers and cooperate in future research if possible.

Reviewer 3 Report

Thank you for response. accept as is

Author Response

Response to Reviewer # 3 Comments Round 2

Comments

Thank you for response. accept as is

Response

The authors are very thankful for the Reviewer positive feedback on our revised manuscript. We really appreciate the acceptance of our paper.

Reviewer 4 Report

Please make changes in the title of the manuscript (pleasу see previous reviewer's comment about this point). P. 11 - delete the term "microstructure". As I noted earlier you didn't investigate MICROstructure, you stidied the structure of the sampples using XRD. It is structure, not microstructure. It is wrong to write this term. 

Author Response

Response to Reviewer # 4 Comments Round 2

Comments

Please make changes in the title of the manuscript (pleasу see previous reviewer's comment about this point). P. 11 - delete the term "microstructure". As I noted earlier you didn't investigate MICROstructure, you stidied the structure of the sampples using XRD. It is structure, not microstructure. It is wrong to write this term.

Response

We thank the reviewer for highlighting this important remark, and we have modified the title to “Structural Study of Nano-clay and its effectiveness in radiation protection against X-rays”

Also, we deleted the words “microstructure and microstructural” from all manusript.

Round 3

Reviewer 2 Report

The revised version of the article is weak, and does not fit the standard level of Nanomaterials.

Basically, authors determined the structure and compositions of a set of natural and synthetic clays by X-ray diffraction, and tested the same materials as X-ray shields. No any clear relation is presented between the nature of the clay and the attenuation performance, only the obvious statement that heavy metals likely adsorb better the X-rays.

In authors response letter, authors continue to stress that they measured attenuation coefficients above 99%. This pure figure is unfortunately not important by its own, since the attenuation depends most importantly on the sample thickness. Any comparison with other reported values from the literature in this paper is therefore meaningless, since in the cited papers, samples of different thicknesses were used. The same is for the used energy range - attenuation is dependent on the X-ray energy.

Furthermore, authors write erroneous statements, that is, the mass adsorption coefficient depends on the sample thickness (page 18)

In their last comment, authors stick to their belief that in one of their samples Astatine element was present, and argue that it disappeared for the time of their second experiment. The probability of such scenario is miserable. Astatine isotopes have half-lives of 8 hours or less, and it is impossible that the initial sample they excavated could contain any measureable amount of it. Although it is only a minor detail in the article, it is also an indicator of the quality of the work.

Author Response

Response to Reviewer 2 Comments Round 3

We have chosen the journal “Nanomaterials” because of its good reputation and high level in the fields related to the synthesis, characterization, and applications of nanomaterials.

But we suspect that the Reviewer has deleted part of our answer to the first point of his/her comment, and we hope that this is purely a mistake and not intentional, especially since most of his comments are negative and lack objectivity and impartiality.

Furthermore, we would like to remind the Reviewer that we have replied to all comments raised by the Reviewers satisfactorily and, as u can see from the revised version of the manuscript, tremendous efforts and extensive improvements are performed by the authors.

Nonetheless, it seems impossible to satisfy the Reviewer 2: each time we respond to his/her comments, he/she bring new comments and his/her comments are very "aggressive and disrespectful".

We therefore cannot tolerate such an attitude and we would like to ask your understanding and kind support to resolve this matter.

We cannot continue this type of discussion with this Reviewer anymore.

We look forward to hearing your positive feedback, knowing that all the other three reviewers submitted a positive report and agreed to publish the article upon two revisions.

We thank you for your cooperation.

Point 1

Basically, authors determined the structure and compositions of a set of natural and synthetic clays by X-ray diffraction, and tested the same materials as X-ray shields. No any clear relation is presented between the nature of the clay and the attenuation performance, only the obvious statement that heavy metals likely adsorb better the X-rays.

In authors response letter, authors continue to stress that they measured attenuation coefficients above 99%. This pure figure is unfortunately not important by its own, since the attenuation depends most importantly on the sample thickness. Any comparison with other reported values from the literature in this paper is therefore meaningless, since in the cited papers, samples of different thicknesses were used. The same is for the used energy range - attenuation is dependent on the X-ray energy.

Response

It is important to highlight that the clay material used in this study is natural, eco-friendly, abundant, and very cheap compared to less abundant, expensive, and toxic metals such as lead or tengesten as reported in the literature.

Gülbiçim, H et al, made a comparison as follows “it doesn’t matter even thought the vermiculite layer of shielding should be thicker than lead or tungsten layer to absorb the same amount of radiation, because the vermiculite has very low density when compared to lead “[3].

[3] Gülbiçim, H, Tufan, MC, Türkan, MN. The investigation of vermiculite as an alternating shielding material for gamma rays. Radiation Physics and Chemistry 130 (2017) 112-117.

As for the comparison made with the results obtained by Vagheian et al. [1], it should be highlighted that the authors used a weak energy in the range 8 - 14 KeV, compared to higher energy used in this study reaching up 30 keV.

In addition, Kim, S.-C. et al. [2] investigated tungsten-based materials with thicknesses in the range of millimeters subjected to different post processing cold and hot rolling. It was found that the obtained results are preferable to lead-based materials, since the highest shielding rate of 100% was achieved for the hot rolled plate with a thickness of 0.3 mm and an energy 24.6 keV.

This reference was added with the correposning explanation, to support the results obtained in this study.

Point 2

Furthermore, authors write erroneous statements, that is, the mass adsorption coefficient depends on the sample thickness (page 18)

Response

Regarding the effect of the sample thickness on the adsorption coefficient, kindly refers to the following references

doi:10.1016/j.biosystemseng.2006.02.013

https://doi.org/10.1016/S0168-9002(99)01257-7

Point 3

In their last comment, authors stick to their belief that in one of their samples Astatine element was present, and argue that it disappeared for the time of their second experiment. The probability of such scenario is miserable. Astatine isotopes have half-lives of 8 hours or less, and it is impossible that the initial sample they excavated could contain any measureable amount of it. Although it is only a minor detail in the article, it is also an indicator of the quality of the work.

Response

Scientific research is based on experience and repetability of the experiment, and some errors may occur during the analysis, and the recognition of the error correction is positive and important in scientific research. After your remarks, we have re-analyzed the samples and the Astatine element was not observed. This may be a mistake in the phases’ autosearch carried out by HighScore program. Kindly refer to the following image from HighScore. You can see that the phase was used in the first Rietveld analysis, as well as some other phases that contain the same error: presence of Astanate element. So it is a systematic error.

Reference code:                            96-900-0147

Mineral name:                                       Vermiculite

Compound name:                                 Vermiculite

Common name:                                    Vermiculite

Chemical formula:                          Mg13.64Si11.44Al4.56O48.00At14.88 

Reference code:                                       96-900-4393

Mineral name:                                          Guilleminite

Compound name:                                    Guilleminite

Common name:                                       Guilleminite

Chemical formula:                                    Ba2.00U6.00Se4.00O28.00At6.00 

Reference code:                            96-900-9473

Mineral name:                                       Gyrolite

Compound name:                                 Gyrolite

Common name:                                    Gyrolite

Chemical formula:    Na1.00Ca16.00Si23.00Al1.00O68.00At14.00
